# Herbicidal Formulations with Plant-Based Compounds to Control *Amaranthus hybridus, Lolium multiflorum*, and *Brassica rapa* Weeds

**DOI:** 10.3390/plants14020276

**Published:** 2025-01-18

**Authors:** Juan J. Romero, Juliana Soler-Arango, Marcos E. Coustet, Daniela B. Moracci, Sebastián Reinoso, Marcos E. Yanniccari, Aline Schneider-Teixeira, Jimena M. Herrera

**Affiliations:** 1YPF Tecnología (Y-TEC), Av. del Petróleo S/N entre 129 y 143, Berisso 1923, Argentina; juanjose.romero@ypftecnologia.com (J.J.R.); juliana.solerarango@ypftecnologia.com (J.S.-A.); marcos.coustet@ypftecnologia.com (M.E.C.); daniela.b.moracci@set.ypf.com (D.B.M.); sebastian.reinoso@ypftecnologia.com (S.R.); aline.s.teixeira@ypftecnologia.com (A.S.-T.); 2Consejo Nacional de Investigaciones Científicas y Técnicas (CONICET), Godoy Cruz 2290, Ciudad Autónoma de Buenos Aires 5000, Argentina; jimenita_herrera@yahoo.com.ar; 3Chacra Experimental Integrada Barrow (MDA—INTA), Ruta 3. Km 487.5, Tres Arroyos 7500, Argentina; 4Instituto Multidisciplinario de Biología Vegetal (CONICET—UNC), Av. Vélez Sarsfield 1611, Córdoba 5016, Argentina

**Keywords:** binary formulations, bioactive compounds, herbicide, natural compounds, weeds

## Abstract

Numerous studies have shown the potential effect of bioactive agents against weeds. In this study, we developed two binary formulations with nonanoic acid, citral, or thymoquinone as herbicides and evaluated their physicochemical properties. The presence of the bioactive compounds in the formulations was confirmed through FTIR spectroscopy. A dynamic light scattering study was conducted to characterize the emulsified formulations and the size and distribution of the aggregates. In addition, thermogravimetric analysis was performed to ensure the thermal stability of the formulations. The herbicidal activity against *Amaranthus hybridus*, *Lolium multiflorum*, and *Brassica rapa* weeds was evaluated, and each species showed different levels of sensitivity with half maximal inhibitory concentration doses from 0.07 to 5 mM. The binary formulations negatively affected the photosynthetic system reducing Fv/Fm values at 5 days after treatment. Lastly, the phytotoxic effect of the formulations was tested on wheat germination, and they did not inhibit plant germination and seedling growth at ≤5 mM after 14 days of application. The development of new formulations with natural compounds as bioactive ingredients would allow control of a wide spectrum of weeds through a multitarget-site effect.

## 1. Introduction

The world population growth projections indicate that higher levels of food production will be required [1]; therefore, agricultural best management practices represent a critical aspect of production systems. *Amaranthus hybridus*, *Lolium multiflorum*, and *Brassica rapa* are common weed species in Argentina and are present in 40% of crop fields [2]. *A. hybridus* is an annual Amaranthaceae weed found in tropical and subtropical regions and produces a large quantity of seeds easily dispersed, *B. rapa* is an annual Brassicaceae weed found in all continents, *L. multiflorum* is an annual weed found in large part of the Pampas region, and the three species have evolved herbicide resistance [2,3,4,5,6,7]. Anyway, the management of these weeds continues to be based on chemical control [2]. However, the increase in herbicide usage has been associated with environmental and human health risks [8]. Currently, some herbicides such as atrazine have been listed as restricted in some countries [9]. An alternative to weed control is the use of plant-based compounds. These are considered of minimum risk due to their low toxicity, easy degradability in the environment, and non-restricted use [9].

Numerous studies have shown the potential effect of botanical agents, such as fatty acids, aldehydes, ketones, terpenes, and others, against weeds [10,11]. The phytotoxic effect of essential oils (EOs) and their pure components have been studied. For example, the herbicidal activity of pure terpenes inhibits the germination and seedling growth in *Sinapis arvensis*, *Amaranthus retroflexus*, *Centaurea solstitialis*, *Raphanus raphanistrum*, *Rumex nepalensis*, and *Sonchus oleraceus*. Kordali et al. [12] and Azirak and Karaman [13] reported that the herbicidal effect of pure terpenes was higher than that of the commercial herbicide 2,4-D isooctyl ester. The effect of these terpenes has been attributed to their interference with plant cell processes, including mitosis inhibition and decreased cellular respiration and chlorophyll content [11].

Nonanoic acid is a saturated fatty acid derived from *Pelargonium* spp. EO, and it is used as a post-emergent weed control agent [14]. Citral (geranial and neral) is an aldehyde terpene found in the EOs of several aromatic plants, such as *Cymbopogon citratus*, *Citrus* sp., and *Lippia* sp. [15,16], and is a weed germination inhibitor [17,18]. Thymoquinone is one of the main active compounds of the EO from black cumin (*Nigella sativa*) and exhibits a wide range of activities such as pharmacological and biopesticide activities [19,20]. These active compounds are also used in the food industry as natural antimicrobials and additives for food preservation [21].

The objective of the current work was to develop binary formulations based on bioactive compounds with herbicidal or biopesticide activities. An emulsifier was used due to the volatility of the compounds [18]. In addition, physicochemical properties and herbicidal activity against weeds were evaluated. Thus, the effect of the binary formulations on PSII was tested. Lastly, the phytotoxic effect of the formulations was evaluated on wheat germination and seedling growth. Therefore, the development of new formulations with different bioactive ingredients would allow control of a wide spectrum of weeds through a multitarget-site effect and new modes of action while causing a low impact on the environment.

## 2. Materials and Methods

### 2.1. Plant Materials

*Amaranthus hybridus* and *Brassica rapa* seeds were collected from weed populations naturalized in crop fields from Tres Arroyos, Argentina (Figure 1A,B). Both species are the most relevant herbicide-resistant weeds in Argentina; they have evolved resistance to multiple herbicides (glyphosate and acetolactate synthase inhibiting herbicides) [2] and frequently reach high density in crop fields (Figure 1A,B). *Lolium multiflorum* was supplied by Barenbrung, Pergamino, Buenos Aires, Argentina. Seeds of *Lactuca sativa* L. var. *crispa* and *Solanum lycopersicum* L. var. *platense* were used as model plants and obtained from Facultad de Ciencias Agrarias y Forestales, Universidad Nacional de La Plata, Argentina. Wheat seeds of var. Tero were provided by Illinois, Argentina.

### 2.2. Chemical Compounds

The bioactive compounds nonanoic acid (96%, catalog number 807167), citral (96%, catalog number 8024890250), and thymoquinone (98%, catalog number 274666) were analytical grade and purchased from Sigma-Aldrich Chemical Co. (Steinheim, Germany) (Table 1). The soy lecithin emulsifier (35% p/p) and atrazine (positive control, 90%) were provided by YPF S.A., Capital Federal, Buenos Aires, Argentina

### 2.3. Formulations

Two binary formulations (F1 and F2) were developed as bioherbicides. F1 contains nonanoic acid and citral (1:1), while F2 contains nonanoic acid and thymoquinone (1:1). Due to the physiochemical properties of the bioactive agents [22] (Table 1), an emulsifier was added.

### 2.4. Chemical Characterization

#### 2.4.1. Fourier Transform Infrared Spectrometry (ATR-FTIR)

A molecular interaction study was conducted using ATR-FTIR. Spectra were recorded with a Nicolet™ iS™10 (Thermo Scientific, Madison, WI, USA). The spectrum of each sample was the average of three successive scans in the 4000–400 cm^−1^ wavenumber range and recorded as the absorbance (calculated as the logarithm of the reflectance reciprocal).

To analyze F1 and F2 binary formulations, a drop was placed on the diamond ATR crystal using a top plate and pressure arm accessories (Smart iTX accessory, Madison, WI, USA). A spectral analysis was performed with the software Omnic version 9 (Thermo Scientific).

#### 2.4.2. Dynamic Light Scattering (DLS) Measurements

The emulsified formulations, size, and size distribution of aggregates were determined through DLS as a function of temperature using a Zetasizer Nano-ZS90 (Malvern Panalytical, Malvern, UK). Samples were properly dispersed in distilled water at room temperature before their analysis to avoid multiple scattering effects. The experiments were performed with emulsified formulations dispersed in distilled water at 25, 35, and 45 °C, typical summer temperatures in Argentina.

#### 2.4.3. Thermogravimetric Analysis (TGA)

To ensure thermal stability, TGA was performed using a TGA Discovery 5500 analyzer (TA Instruments, New Castle, DE, USA), under nitrogen flux (20 mL min^−1^), in platinum pans heated up to 150 °C with a 10 °C min^−1^ rate and isothermal steps at 25, 35, and 45 °C, typical summer temperatures in Argentina.

### 2.5. Herbicide Activity

#### 2.5.1. Weed Germination Inhibition Test

The herbicidal activity of F1 and F2 was tested against weed and model plant seeds following the methodology described by Sosa et al. [23] with some modifications. Briefly, 4 mL of aqueous emulsions containing F1 or F2 at 0.07 to 5 mM were tested, and an emulsifier was added to the formulation at 0.2% *v*/*v* [24]. The extracts were placed onto 9 cm diameter paper disks in Petri dishes. Then, 10 seeds of each species were placed onto paper disks. Subsequently, the dishes were closed under the following experimental conditions: room temperature, 26.0 ± 1.8 °C; relative humidity, 50 ± 9.7%; and photoperiod, 12:12. At 7 days after sowing, the seeds were considered germinated if their roots were longer than 1 mm [25]. Emulsions without the addition of the bioactive formulation were used as a negative control, whereas atrazine was used as a positive control because it targets a broad spectrum of weeds (monocotyledons and dicotyledons) [24,26]. The assays were performed in triplicates for each concentration. Half maximal inhibitory concentrations (IC50) were determined after 7 days of exposure and were calculated using POLO PLUS 2002–2007 LeOra Software [27].

#### 2.5.2. Effect of Binary Formulations on Photosystem II (PSII)

The damage on PSII of the formulations was evaluated on seedlings of *A. hybridus*, because it is a main weed in a wide variety of crops worldwide [28], following the methodology described by Pooja et al. [29] with modifications. Briefly, 25 seedlings with 7 days post-emergence were placed onto paper filters into plastic trays (15.5 cm × 11 cm × 4 cm). Then, 10 mL of aqueous emulsions containing F1 or F2 at 0.07 to 5 mM and an emulsifier incorporated into the extracts at 0.2% *v*/*v* were added into boxes. Aqueous extracts without the addition of the bioactive formulation were used as a negative control, whereas atrazine (5 mM) was used as a positive control. In order to avoid possible chlorosis symptoms due to a lack of nutrients, 5 mL of nutritive solution (5.0 mM Ca (NO_3_) 2, 5.0 mM KNO_3_, 2.0 mM MgSO_4_, 1.0 mM KH_2_PO_4_, 20.0 μM FeNa EDTA, 5.0 μM H_3_BO_3_, 0.9 μM MnCl_2_, 0.8 μM ZnCl_2_, 0.3 μM CuSO_4_ y 0.01 μM Na_2_MoO_4_, pH 5.5–6.5) was added to each tray. The boxes were placed in a culture chamber at 22 ± 2 °C and 50 μmol photons m^−2^ s^−1^ with a photoperiod of 16 h. The chlorophyll fluorescence measures (Fv/Fm) were carried out with a MINI-PAM II ©Walz fluorimeter (Effeltrich, Germany) at 1, 2, 5, 7, and 12 days after application. The seedlings were darkened 30 min before measurements were taken.

For the experiments, three repetitions per treatment were used: Both experiments had four treatments. Each experiment was replicated three times. The treatments were assigned randomly to each box following a completely randomized design. The statistical analysis for Fv/Fm data was performed using InfoStat 2008 Software through Generalized Mixed Linear Models [30].

#### 2.5.3. Effect of Binary Formulations on Wheat Germination and Seedling Growth

Effects of F1 and F2 on wheat germination and seedling growth were evaluated. The experiments were conducted according to Peschiutta et al. [31] with some modifications. F1 and F2 were added to Petri dishes at 2 and 5 mM and after 1, 7, or 14 days. Ten wheat seeds were placed in the Petri dishes. A negative control (H_2_O) treatment was performed without the addition of any active compound, and the seeds were placed onto paper disks at the same time intervals. Then, the dishes were closed under the same experimental conditions mentioned above. The number of germinated seeds per dish was recorded 7 days after sowing. The seeds were considered germinated if their roots were longer than 1 mm. At that time, the leaf length of the wheat seedlings was measured. The assay was performed in triplicates. Data were analyzed to assess normality using the Shapiro–Wilk test, and homogeneity of the variances was determined using Levene’s test before performing ANOVA. Tukey’s tests were used to compare the means for germination and leaf length of seedlings between treatments through Infostat Software [30].

## 3. Results

### 3.1. Chemical Characterization

#### 3.1.1. Fourier Transform Infrared Spectrometry (ATR-FTIR)

Figure 2 and Figure 3 show the typical FTIR spectra of components corresponding to herbicidal formulations called F1 and F2, respectively. The characteristic band at 3343 cm^−1^ (grey area) that appears for all formulations is attributed to the O-H stretching vibration from the nonanoic acid and emulsifier. In addition, another band from the emulsifier is located at 3008 cm^−1^ (blue area) for F1 (Figure 2), related to asymmetric and symmetric C-H stretching vibrations, whereas the band for F2 has two origins, emulsifier and the thymoquinone, related to =C-H stretching vibrations (Figure 3). The spectrum shows the characteristic bands at 2923.2 cm^−1^ (orange area) associated with the C-H ester stretching vibration, the band at 1675.5 cm^−1^ (red area) is due to the stretching vibration of the ketone and aldehyde C=O group, and the band at 1117 cm^−1^ (green area) is attributed to the ether (C-O) groups. The last three bands mentioned are present in all the F1 and F2 components.

#### 3.1.2. Dynamic Light Scattering (DLS) Measurements

To characterize the emulsified formulations, hydrodynamic diameters were determined for each sample through DLS. F1 and F2 showed unimodal size distributions, whereas a multimodal distribution was observed for soy lecithin emulsifier (Figure 4A). Thus, for the temperature range involved in the experiments, the average sizes were from 800–1300 nm for the emulsifier, while F1 and F2 showed dispersion sizes of 250–550 and 170–260 nm, respectively.

The presence of nonpolar analytes (active compounds) in the formulation turned the E dispersions into clear emulsions with unimodal distributions and lower polydispersity indexes. In contrast, the observed change in F1 and F2 size distributions with temperature was in line with the expected values for stable colloidal systems, with no flocculating aggregates under our experimental conditions (Figure 4B).

#### 3.1.3. Thermogravimetric Analysis (TGA)

TGA was used to assess the thermal stability of the formulations. Thermograms of pure active compounds (thymoquinone, citral, and nonanoic acid), emulsifier, F1, and F2 were obtained for comparison. Weight losses of less than 10% were observed at temperatures under 100 °C for pure active compounds, except thymoquinone, which lost 17.5% of its weight. According to previously reported thermal assays (Figure 5A), this behavior can be the result of evaporation processes, considering the boiling point of these compounds and the constant N_2_ flux (Table 1). In contrast, Figure 5A shows that the formulation composition did not affect significantly the recorded weight losses. In these cases, the presence of water was responsible for the considerable weight losses at temperatures under 100 °C, and the final weight loss corresponded to the remaining amount of the sample, consisting of active compounds and the emulsifier in the formulations. Lastly, the active compounds showed adequate thermal stability at all temperatures tested.

### 3.2. Herbicide Activity

#### 3.2.1. Weed Germination Inhibition Test

Germination inhibition of weeds induced by F1 and F2 formulations is shown in Table 2. F1 and F2 caused phytotoxic effects on all weeds; however, *A. hybridus* and *L. multiflorum* species were the most sensitive. F1 was more active than F2 for all weeds tested, and both formulations showed greater herbicidal activity than atrazine (positive control).

#### 3.2.2. Effect of Binary Formulations on Photosystem II (PSII)

The effect of formulations F1 and F2 on PS II at 5 mM is shown in Figure 6. At a concentration lower than 5 mM, no significant differences among treatments were found. At 1 and 2 days after application, Fv/Fm values were lower for F1 and F2 compared to the controls. At 5 days after treatments, the Fv/Fm was zero due to the seedlings exhibiting severe damage symptoms in response to F1 and F2. However, the seedlings’ exposure to atrazine showed a gradual decline in values of Fv/Fm from 5 to 12 days after treatment. In contrast, the control without herbicide did not show damage to PSII during the period evaluated (Figure 6).

#### 3.2.3. Effect of Binary Formulations on Wheat Germination and Seedling Growth

The effect of F1 and F2 on wheat germination and seedling growth is shown in Figure 7 and Figure 8. F1 and F2 were toxic 1 day after application for both concentrations. However, at 7 and 14 days after application, no significant differences were observed in the germination percentages for F1 at 2 mM and F1 at 5 mM compared to the control. Both formulations applied at 2 and 5 mM did not affect wheat germination 14 days after application (Figure 7A,B). Regarding leaf growth, both F1 and F2 at 5 mM produced growth inhibition at 7 and 14 days after application. Less growth was observed 7 days after application for both formulations at 2 mM, and no significant difference was observed 14 days after application (Figure 8A,B).

## 4. Discussion

### 4.1. Chemical Characterization

The presence of active compounds and the emulsifier in herbicidal formulations (F1, F2) was confirmed by FTIR spectroscopy through the databases [32,33,34,35].

These emulsified formulations showed narrow dispersion sizes evidencing their stability between 25 and 45 °C; however, a multimodal distribution was observed for emulsifiers through DLS measurements. Current results are consistent with the complex chemical nature of soy lecithin emulsifier, which is composed of a vast mixture of amphiphilic compounds of diverse structures, mainly phospholipids and triglycerides, enabling different assembly options [32,33,34,35,36,37]. Soy lecithin has been successfully used as a natural emulsifier to improve the accessibility of different lipophilic nutrients or bioactives, showing a similar behavior [38]. Emulsification of active compounds is a key strategy in the food, cosmetic, and pharmaceutical industries [39]. The current results obtained from F1 and F2 suggest microphase reorganization into new colloidal arrangements [40]. The goal of this approach is to form stable dispersions of nonpolar additives in a polar medium (it can also be used in the opposite way) that can be properly dozed in commercial products [41].

TGA showed adequate thermal stability for nonanoic acid, citral, and thymoquinone in F1 or F2 at all temperatures tested. From these experiments, it can be observed that decomposition profiles depended on the chemical nature of active molecules; for example, citral and thymoquinone are volatile compounds and evaporate at a temperature between 45 and 165 °C, while nonanoic acid evaporates at 100 °C and above [33,42,43]. Despite ambient conditions involving the presence of atmospheric O_2_, our experiments were conducted under N_2_ flux. At high temperatures, oxygen may react with active compounds [44], but at temperatures tested, the oxidation reactions with small amounts of atmospheric oxygen should have a minimum impact; therefore, our results can be considered a useful approximation to actual thermal stability [33,42,43].

### 4.2. Herbicide Activity

The herbicide activity of F1 and F2 was evidenced on *A. hybridus*, *B. rapa*, *L. multiflorum*, *L. sativa*, and *S. lycopersicum* during the germination process, and IC50 values were from 0.07 to 4.6 mM according to the bioactive formulation and species (Table 2). F1 was more active than F2 in *B. rapa* and *L. sativa*. No differences between formulations were detected in the sensitivity of *A. hybridus*, *L. multiflorum*, and *S. lycopersicum* to both formulations. Previous studies reported by Sosa et al. [23] showed the action of plant-derived bioactive molecules on ryegrass (*L. multiflorum*) and lettuce (*L. sativa*) with IC50 of 0.6 and 0.7 mM, respectively. In agreement with our results, Dudai et al. [17] found different levels of sensitivity to citral obtained from *Cymbopogon citratus* EO (42.6% geranial and 32.1% neral) depending on the plant species (IC50 from 0.00008 to 0.000116 mM).

F1 was prepared with two bioactive agents, citral and nonanoic acid, and previous research reported that citral reduces cell division, disrupts mitotic microtubules and cell plates, and inhibits cell elongation by damaging cortical microtubules, whereas nonanoic acid causes loss of membrane integrity and rapid cell death [18,44,45,46]. Meanwhile, F2 consists of nonanoic acid and thymoquinone, and the mechanism of action of thymoquinone as an herbicide has not been determined. Herrera et al. [19] have reported insecticidal activity of this bioactive through acetylcholinesterase inhibition. The structure–activity research showed that topological and/or physicochemical properties can be related to the biological effects on pests. Herrera et al. [47] proposed that electronic descriptors of terpenes, such as the orbital electronegativity of the carbonyl group, are associated with enzyme inhibition in pests. Hence, F1 and F2 may be developed as promising herbicides against weeds and used in organic farming systems.

Current work demonstrates that the injuries caused by F1 and F2 on the photosynthetic apparatus of *A. hybridus* were greater than those caused by atrazine treatment (Figure 6). The measurement of Fv/Fm has been used to estimate the damage to the PSII [29,48,49]. In non-senescent and mature leaves, the values of Fv/Fm are about 0.8 [50]. For the genus *Amaranthus*, values of 0.72 were found for unstressed leaves [51]. In this study, the maximum values of Fv/Fm for the control treatment were about 0.7, and it was 0.1 1 day after the F1 or F2 treatment (Figure 6). Concerning the effects of EOs and their pure compounds on plant physiology, several studies reported a reduction in the chlorophyll content [11,18,52]. The studies of fluorescence emission analysis of photosynthetic apparatus have been used to determine the mode of action of new herbicides [53,54].

Several studies found that some EO compounds can be phytotoxic against crops [55,56]; however, crop selectivity is a desirable trait in the development of herbicides, and the effects of F1 and F2 on wheat germination and seedling growth were approached in the current work. Both formulations were tested in pre-planting treatments, and they did not affect wheat germination when the sowing took place 7 or 14 days after application, depending on the formulation and concentration (Figure 7A,B). Regarding leaf growth, both F1 and F2 at 5 mM produced growth inhibition of wheat seedlings at 7 and 14 days after application. Less growth was observed at 7 days after application for both formulations at 2 mM, and no significant differences were observed at 14 days after application (Figure 8A,B). It is necessary to know the EO potential effect on food crops to determine the planting time at which phytotoxicity does not affect the crop. In this sense, Synowiec et al. [57] reported that caraway or peppermint EOs can selectively inhibit the growth of *Echinochloa crusgalli* but not that of maize plants. Ibañez and Blázquez [58] pointed out that oregano EO is the most harmful for cucumber and tomato seedlings, whereas rosemary EO is the least harmful for seed germination in these seedlings. This study, conducted on a laboratory scale, showed the stability and persistence of the formulations over time. We determined that after 14 days of application, the formulations did not inhibit wheat germination or growth of seedlings.

## 5. Conclusions

Two herbicidal formulations were developed on plant-based compounds. We demonstrated that the functional groups of the active compounds in both binary formulations were maintained, showing unimodal size distributions and thermal stability at the temperatures tested. The formulations were non-selective and controlled a broad spectrum of species (monocotyledons and dicotyledons); however, *A. hybridus* and *L. multiflorum* were the most sensitive. The binary formulations of nonanoic acid/citral and nonanoic acid/thymoquinone negatively affected the photosynthetic system of *A. hybridus*. The phytotoxic effect of the formulations was tested on wheat germination, and they did not inhibit plant germination and seedling growth after 14 days of application. However, further research should consider the validation of the effectiveness of F1 and F2 at the field scale. New herbicide formulations developed with different bioactive ingredients would control a wide spectrum of weeds through multitarget-site effects, new modes of action, and a low impact on the environment.

## Figures and Tables

**Figure 1 plants-14-00276-f001:**
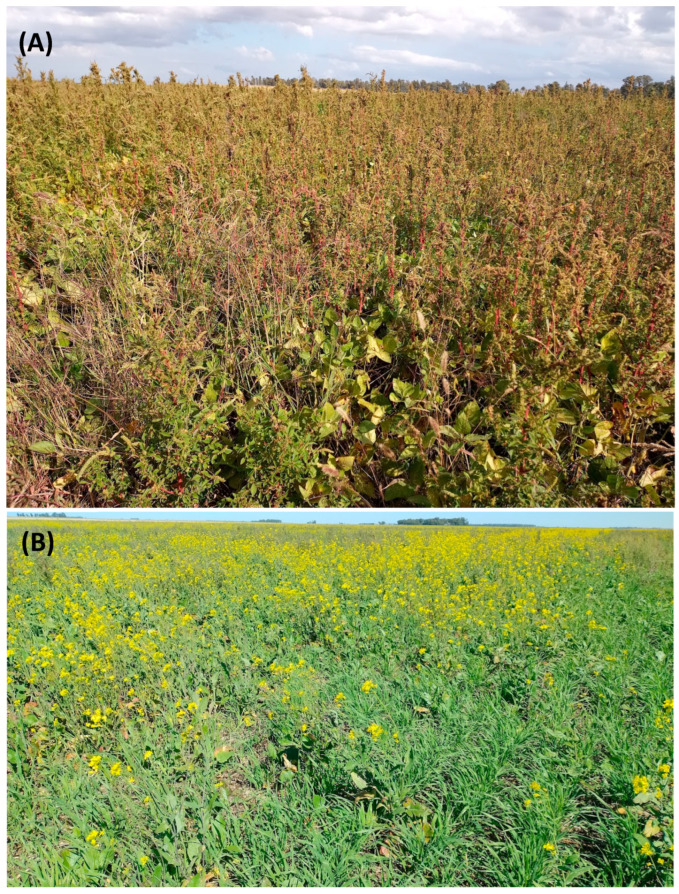
Multiple herbicide-resistant weed populations harvested to obtain seeds employed in the experiments: (**A**) *Amaranthus hybridus* in soybean crop and (**B**) *Brassica rapa* in oat crop.

**Figure 2 plants-14-00276-f002:**
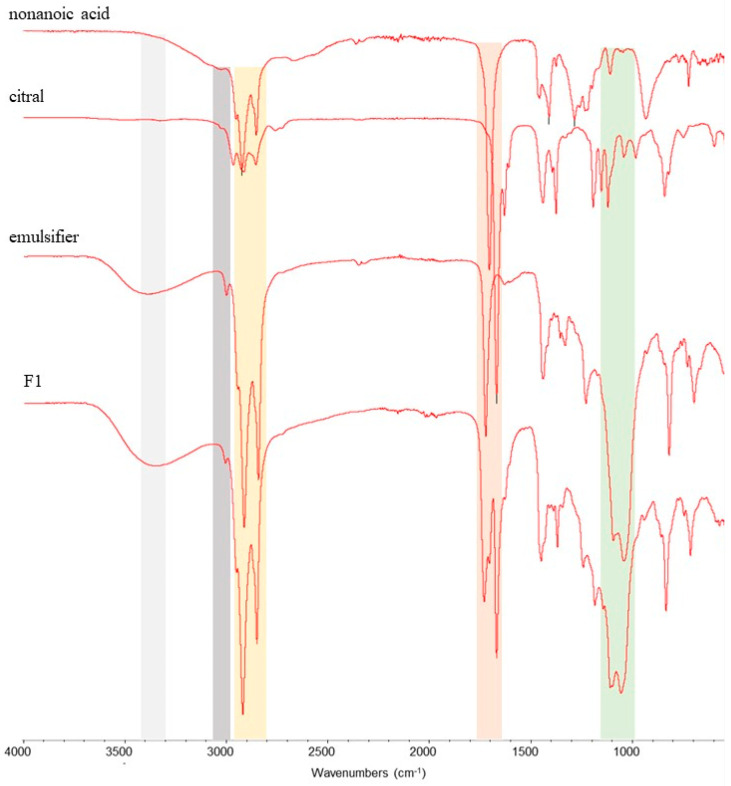
FTIR spectra of components (nonanoic acid, citral, and emulsifier) corresponding to formulation F1. The colored areas represent the bands of the oxygenated organic compound functional groups of interest: grey, OH; grey, =C-H; orange, C-H; red, C=O; and green, C-O.

**Figure 3 plants-14-00276-f003:**
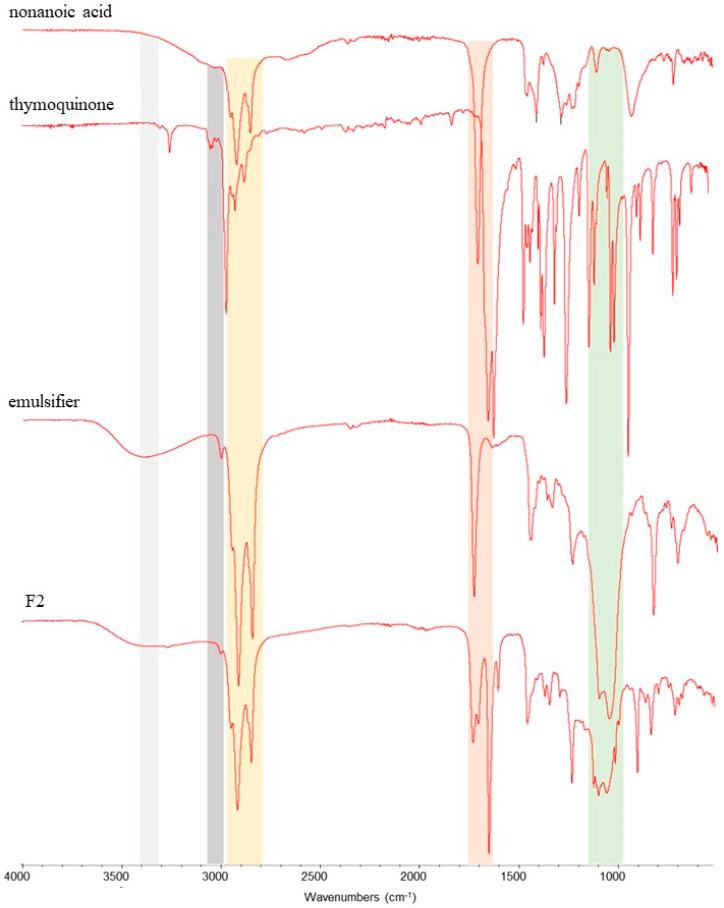
FTIR spectra of components (nonanoic acid, thymoquinone, and emulsifier) corresponding to formulation F2. The colored areas represent the bands of the oxygenated organic compound functional groups of interest: grey, OH; grey, =C-H; orange, C-H: red, C=O; and green, C-O.

**Figure 4 plants-14-00276-f004:**
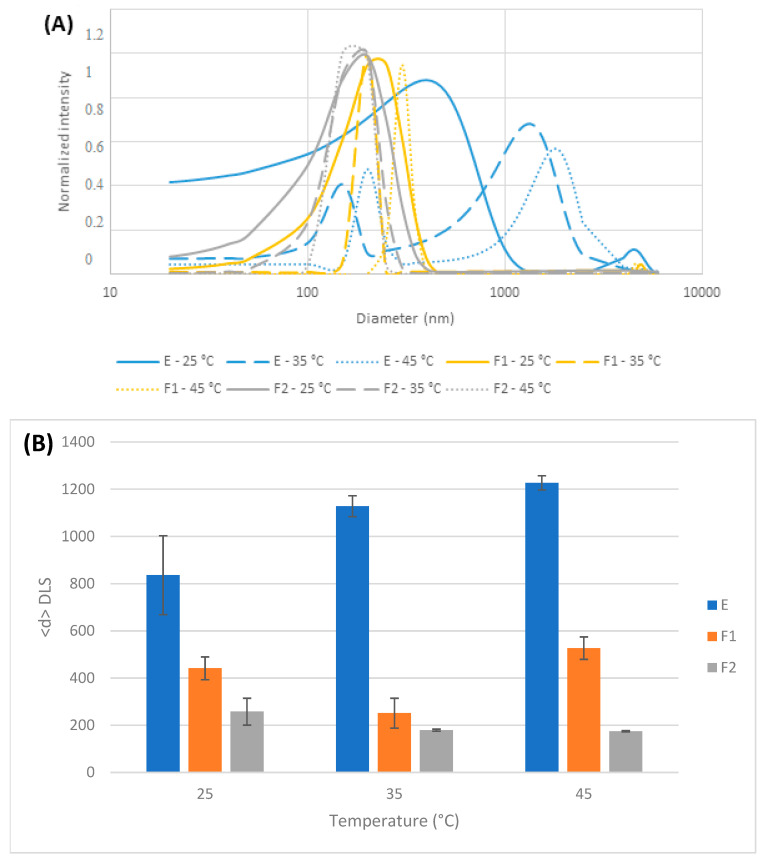
(**A**) DLS size distribution for emulsifier (E, blue), formulation 1 (F1, orange), and formulation 2 (F2, grey) at 25 (line), 35 (slashed), and 45 °C (dotted). (**B**) Temperature effect on averaged diameter (d) for emulsifier (E, blue), F1 (orange), and F2 (grey).

**Figure 5 plants-14-00276-f005:**
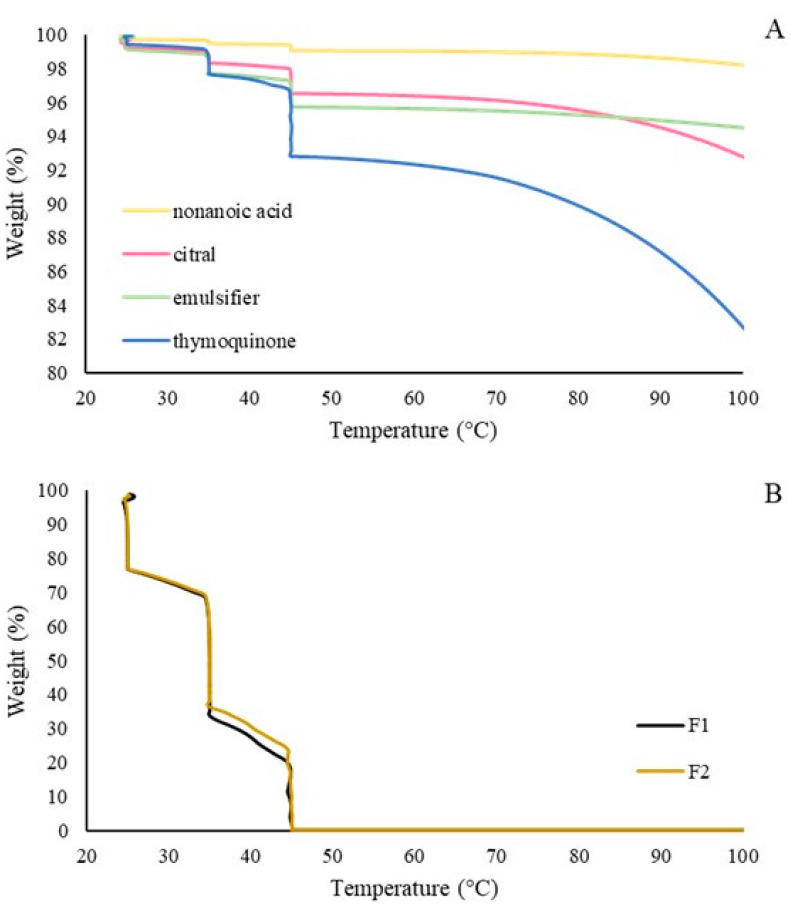
TGA thermograms: (**A**) pure active compounds (nonanoic acid, citral, thymoquinone, and emulsifier) and (**B**) formulations (F1 and F2).

**Figure 6 plants-14-00276-f006:**
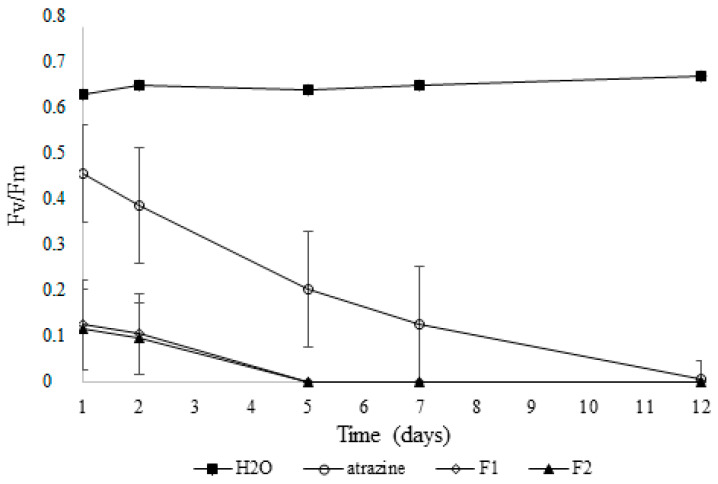
Effect of binary formulation F1 and F2 on PSII on seedling of *A. hybridus*.

**Figure 7 plants-14-00276-f007:**
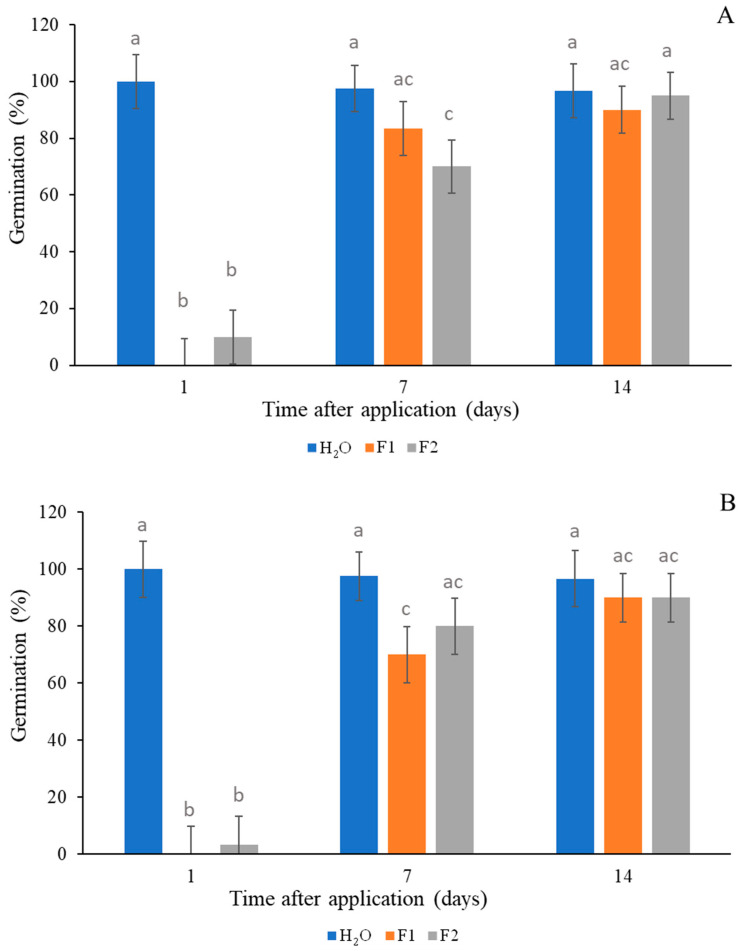
Percentage of germination of wheat seeds in response to different treatments (F1, orange; F2, grey and H_2_O, blue) at 2 mM (**A**) and 5 mM (**B**) at 1, 7, and 14 days after application. Different letters indicate significant differences (*p* < 0.05).

**Figure 8 plants-14-00276-f008:**
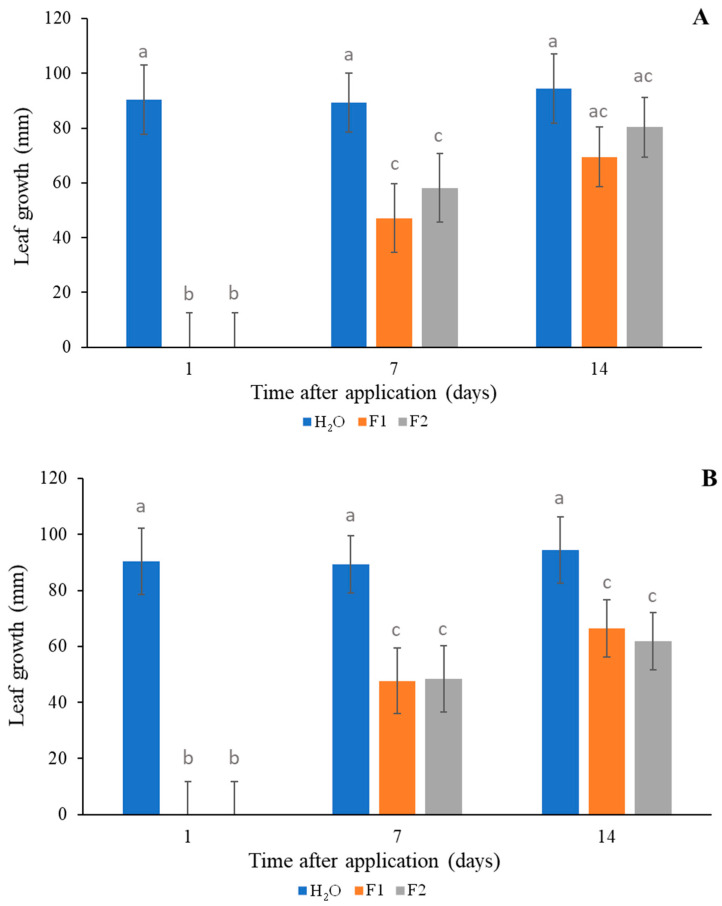
Leaf growth of wheat seedlings in response to different treatments (F1, orange; F2, grey and H_2_O, blue) at 2 mM (**A**) and 5 mM (**B**) at 1, 7, and 14 days after application. Different letters indicate significant differences (*p* < 0.05).

**Table 1 plants-14-00276-t001:** Bioactive compounds and physicochemical properties.

Chemical Structures	Boiling Point (°C)	Log P	Solubility in Water at 25 °C (g/L)
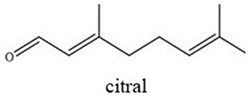	225	3.17	0.08
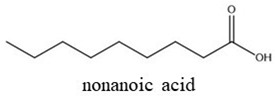	254	3.43	0.21
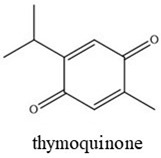	230–232	2.33	0.87

**Table 2 plants-14-00276-t002:** Germination inhibition of seeds after 7 days of exposure to bioactive formulations.

Treatments	*A. hybridus*	*B. rapa*	*L. multiflorum*	*L. sativa*	*S. lycopersicum*
	IC_50_ (mM) ^1^ (95% Confidence Interval)				
F1	0.07 (0.03–0.09) **^a^**	1.5 (1.3–1.7) **^a^**	0.24 (0.09–0.319) **^a^**	2.54 (2.21–2.9) **^a^**	1.33 (0.89–1.63) **^a^**
F2	0.07 (0.04–0.08) **^a^**	2.3 (1.7–2.9) **^b^**	0.14 (0.08–0.27) **^a^**	4.60 (3.36–10.00) **^b^**	0.99 (0.37–1.38) **^a^**
Atrazine	41.0 (25.1–54.1) **^c^**	300 (189.1–368.8) **^c^**	38.9 (25.1–56.1) **^c^**	383.5 (307.1–454.8) **^c^**	37.5 (15.3–49.1) **^c^**

^1^ Inhibition concentration (IC_50_) values between treatments were considered significantly different if their confidence limits did not overlap. Different letters indicate significant differences. The experiment was performed in triplicates.

## Data Availability

Data are available from the authors upon reasonable request. The data are not publicly available due to privacy.

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
