# Peer review of "Herbicidal Formulations with Plant-Based Compounds to Control Amaranthus hybridus, Lolium multiflorum, and Brassica rapa Weeds"

_plants, 2025, doi:10.3390/plants14020276_

Round 1
Reviewer 1 Report
Comments and Suggestions for Authors
In this manuscript, Juan J. Romero and colleagues have developed two binary formulations with nonanoic acid, citral, or thymoquinone as herbicides and evaluated their physicochemical properties. This study is impressive and with practical interests, and I have the following comments:
1, For the Title, I suggest to employ “Herbicidal formulations with plant-based compounds to control Amaranthus hybridus, Lolium multiflorum, and Brassica rapa weeds”.
2, For the Abstract, please consider to remove “new modes of action, and with a low impact on the environment.”
3, For the Keywords, “photosystem II” should be deleted.
4, For the Introduction, the impact of results from this study on agriculture should be stated in the last paragraph.
5, For the Materials and methods, detailed information for weed population of Amaranthus hybridus and Brassica rapa examined in this study should be described. Pictures showing the growth of weeds in field is appealing. Commercial catalog of bioactive compounds nonanoic acid, citral, and thymoquinone should be provided.
6, For the Results, a summarizing title should be provided for Figure 3 legend. The abbreviations appeared in the Figures like <d> DLS (Y-axis in the Figure 3b) should be explained. Standard values should be listed in the Table 2. In addition, please use emulsifier instead of E throughout the manuscript, since its abbreviation E is confusing.
7, For the Discussion, authors should consider to divide this section into several subsections and each subsection should be entitled.
8, For the Conclusion, please combine the two paragraphs into one paragraph.
9, Provisional patent does not need.
Author Response
Thank you for your comments and suggestions. The changes were highlighted in yellow.
Comments and Suggestions for Authors
In this manuscript, Juan J. Romero and colleagues have developed two binary formulations with nonanoic acid, citral, or thymoquinone as herbicides and evaluated their physicochemical properties. This study is impressive and with practical interests, and I have the following comments:
1, For the Title, I suggest to employ “Herbicidal formulations with plant-based compounds to control Amaranthus hybridus, Lolium multiflorum, and Brassica rapa weeds”.
Response: The title was modified.
2, For the Abstract, please consider to remove “new modes of action, and with a low impact on the environment.”
Response: Lines 29-30. The sentence was deleted.
3, For the Keywords, “photosystem II” should be deleted.
Response: Done.
4, For the Introduction, the impact of results from this study on agriculture should be stated in the last paragraph.
Response: Done.
5, For the Materials and methods, detailed information for weed population of Amaranthus hybridus and Brassica rapa examined in this study should be described. Pictures showing the growth of weeds in field is appealing. Commercial catalog of bioactive compounds nonanoic acid, citral, and thymoquinone should be provided.
Response: Information about weed populations and pictures were added. The bioactive compounds nonanoic acid (96%), citral (96%), and thymoquinone (98%) were analytical grade and purchased from Sigma-Aldrich Chemical Co. The catalog numbers were added in the text.
6, For the Results, a summarizing title should be provided for Figure 3 legend.
The abbreviations appeared in the Figures like <d> DLS (Y-axis in the Figure 3b) should be explained. Standard values should be listed in the Table 2. In addition, please use emulsifier instead of E throughout the manuscript, since its abbreviation E is confusing.
Response: Figure 3 (it is named Figure 4, a new figure was added as Fig 1) and figure legend were modified. The abbreviation “E” for emulsifier was remove in all manuscript, except, in the Figure 4 for graphical format.
Table 2 showed the values of inhibition concentration (IC50) and their confidence limits confidence calculated through POLO PLUS Software. Here, they were considered significantly different if their confidence limits did not overlap. Table legend was clarified.
7, For the Discussion, authors should consider to divide this section into several subsections and each subsection should be entitled.
Response: the discussion was divided into two sections.
8, For the Conclusion, please combine the two paragraphs into one paragraph.
Response: Done
9, Provisional patent does not need.
Response: The section was deleted.
Reviewer 2 Report
Comments and Suggestions for Authors
The manuscript is focused on the development of novel formulations of bioherbicides against Amaranthus hybridus, Lolium multiflorum, and Brassica rapa weeds. The topic is of practical interest and of high demand for agriculture. The idea to decrease toxicity and increase degradability by application of natural compounds with herbicide activity is in current trend in the field. The combination of natural components of plant origin i.e. nonanoic acid, citral, or thymoquinone as herbicides has been prepared and tested as potential bioherbicides. The results obtained allows to conclude that the idea was correct and combinations are effective. Novel herbicide formulations developed with different bioactive ingredients would control a wide spectrum of weeds through multitarget-site effects, new modes of action, and with a low impact on the environment.
The manuscript shows scientific novelty in the idea and experimental data. The experiment is logically designed and explained in details allowing understand the procedures applied. Results and discussion are meaningful. A good comparison of the data obtained to the reported examples for other bioherbicides is presented. Conclusions are supported with the experimental data. Further development ways are pointed out. The reference list is actual and is relevant to the topic of the study.
The manuscript can be accepted to publication after revision. The remarks are listed below.
1. Reference to Figure 7 and corresponding explanation are missed in the text of the manuscript (Section 3.2.3).
2. All figures throughout the manuscript, the statistical difference marked with a, b, c has to be explained in the caption (which parameters are compared in each case).
3. Figures 3A and 5, Y axis, decimal points to be divided with dot but not comma.
Author Response
Thank you for comments and suggestions. The changes were highlighted in green.
Comments and Suggestions for Authors
The manuscript is focused on the development of novel formulations of bioherbicides against Amaranthus hybridus, Lolium multiflorum, and Brassica rapa weeds. The topic is of practical interest and of high demand for agriculture. The idea to decrease toxicity and increase degradability by application of natural compounds with herbicide activity is in current trend in the field. The
combination of natural components of plant origin i.e. nonanoic acid, citral, or thymoquinone as herbicides has been prepared and tested as potential bioherbicides. The results obtained allows to conclude that the idea was correct and combinations are effective. Novel herbicide formulations developed with different bioactive ingredients would control a wide spectrum of weeds through multitarget-site effects, new modes of action, and with a low impact on the environment.
The manuscript shows scientific novelty in the idea and experimental data. The experiment is logically designed and explained in details allowing understand the procedures applied. Results and discussion are meaningful. A good comparison of the data obtained to the reported examples for other bioherbicides is presented. Conclusions are supported with the experimental data. Further development ways are pointed out. The reference list is actual and is relevant to the topic of the study.
The manuscript can be accepted to publication after revision. The remarks are listed below.
- Reference to Figure 7 and corresponding explanation are missed in the text of the manuscript (Section 3.2.3).
Response: Thank you for the observation. The paragraph was added.
- All figures throughout the manuscript, the statistical difference marked with a, b, c has to be explained in the caption (which parameters are compared in each case).
Response: Done.
- Figures 3A and 5, Y axis, decimal points to be divided with dot but not comma.
Response: Done
Reviewer 3 Report
Comments and Suggestions for Authors
The manuscript with the title “Herbicidal formulations with plant-based compounds to control Amaranthus hybridus, Lolium multiflorum, and Brassica rapa” assessed the effect of bioactive agents (nonanoic acid, citral, or thymoquinone) against three common annual weeds (two dicots and one monocot), at concentration 0.07 to 5 mM. In addition, herbicide formulations were tested on wheat germination, in the perspective of using these herbicides in wheat crops. This research is very important because current trend is to search for minimum risk - low toxicity herbicides.
Please check all the scientific botanical names to be correct in accordance with some reputable botanical data base such as IPNI. E.g Line 53 Centaurea salsotitialis shouldn’t it be Centaurea solstitialis? Please check all the scientific names in the article to be correctly spelled.
Line 81 “crispa” and Line 82 “platense” with italics
Line 167 – what software was used for statistical tests?
Material and Method – I suggest authors to consider making a figure as an illustrative flow-chart with pictograms to present very clearly the experimental layout, it would be very attractive and helpful for readers.
Results
ART-FTIR band and peak assignment to function groups from the compounds, was conducted based on comparison with references or databases? Because there is no source for interpretation of bands.
Figures 6 and 7, it is not clear to me how letter-case significance assignment was done. For each application day (1, 7, 14) separately or among all variants at once. (Note: this is why I suggested at Material and method to add a flow chart of the experiment – to be very clear the experimental factors, gradients/levels and the number of resulting variants). Either way, “a” was assigned to highest value as it is indeed usually done, but “b” should be assigned the next highest value which is signifiantly different from a, instead the intermediate value is assigned “c” and the lowest “b”? The figure caption is not very clear either. Please clarify as best as possible for the readers.
Conclusions
According to the last paragraph of the introduction, the aim (general objective) was to create binary formulations and asses their effectiveness. The specific objectives given by authors were to test: 1) volatility of the compounds, 2) physicochemical properties and 3) herbicidal activity on weeds (PSII), 4) effect on crop plant germination and growth (wheat). Hence, I suggest the conclusions to mirror this structure and respond to each specific objective in the conclusions, preferably in the same order.
Best regards.
Author Response
Thank you for comments and suggestions. The changes were highlighted in violet.
Comments and Suggestions for Authors
The manuscript with the title “Herbicidal formulations with plant-based compounds to control Amaranthus hybridus, Lolium multiflorum, and Brassica rapa” assessed the effect of bioactive agents (nonanoic acid, citral, or thymoquinone) against three common annual weeds (two dicots and one monocot), at concentration 0.07 to 5 mM. In addition, herbicide formulations were tested on wheat germination, in the perspective of using these herbicides in wheat crops. This research is very important because current trend is to search for minimum risk - low toxicity herbicides.
Please check all the scientific botanical names to be correct in accordance with some reputable botanical data base such as IPNI. E.g Line 53 Centaurea salsotitialis shouldn’t it be Centaurea solstitialis? Please check all the scientific names in the article to be correctly spelled.
Response: The scientific botanical names of plants were checked and corrected.
Line 81 “crispa” and Line 82 “platense” with italics
Response: Done.
Line 167 – what software was used for statistical tests?
Response: The software used was added in the sentence.
Material and Method – I suggest authors to consider making a figure as an illustrative flow-chart with pictograms to present very clearly the experimental layout, it would be very attractive and helpful for readers.
Response: Thank you so much for your suggestion, a graphical abstract was built.
Results
ART-FTIR band and peak assignment to function groups from the compounds, was conducted based on comparison with references or databases? Because there is no source for interpretation of bands.
Response: Yes, the presence of active compounds and emulsifier was confirmed by FTIR (Line: 301)
Figures 6 and 7, it is not clear to me how letter-case significance assignment was done. For each application day (1, 7, 14) separately or among all variants at once. (Note: this is why I suggested at Material and method to add a flow chart of the experiment – to be very clear the experimental factors, gradients/levels and the number of resulting variants). Either way, “a” was assigned to highest value as it is indeed usually done, but “b” should be assigned the next highest value which is signifiantly different from a, instead the intermediate value is assigned “c” and the lowest “b”? The figure caption is not very clear either. Please clarify as best as possible for the readers.
Response: In section 2.4.3 was clarified. In addition, figure legends were modified.
Conclusions
According to the last paragraph of the introduction, the aim (general objective) was to create binary formulations and asses their effectiveness. The specific objectives given by authors were to test: 1) volatility of the compounds, 2) physicochemical properties and 3) herbicidal activity on weeds (PSII), 4) effect on crop plant germination and growth (wheat). Hence, I suggest the conclusions to mirror this structure and respond to each specific objective in the conclusions, preferably in the same order.
Response: The conclusion was modified following the suggestion.
Round 2
Reviewer 1 Report
Comments and Suggestions for Authors
Authors have addressed my concerns in the revision.